# The Role of Culture and Religion on Sexual and Reproductive Health Indicators and Help-Seeking Attitudes amongst 1.5 Generation Migrants in Australia: A Quantitative Pilot Study

**DOI:** 10.3390/ijerph18031341

**Published:** 2021-02-02

**Authors:** Tinashe Dune, David Ayika, Jack Thepsourinthone, Virginia Mapedzahama, Zelalem Mengesha

**Affiliations:** 1School of Health Sciences, Western Sydney University, Penrith, NSW 2751, Australia; d.ayika@westernsydney.edu.au (D.A.); j.thepsourinthone@westernsydney.edu.au (J.T.); mapedzav@gmail.com (V.M.); 2Translational Health Research Institute, Western Sydney University, Penrith, NSW 2751, Australia; z.mengesha@westernsydney.edu.au; 3Susan Wakil School of Nursing and Midwifery, University of Sydney, Camperdown, NSW 2050, Australia; 4Uniting Care, North Parramatta, NSW 2151, Australia

**Keywords:** 1.5 generation migrants, sexual and reproductive health, Australia, cross-cultural, religiosity

## Abstract

In Australia, 1.5 generation migrants (those who migrated as children) often enter a new cultural and religious environment, with its own set of constructs of sexual and reproductive health (SRH), at a crucial time in their psychosexual development—puberty/adolescence. Therefore, 1.5 generation migrants may thus have to contend with constructions of SRH from at least two cultures which may be at conflict on the matter. This study was designed to investigate the role of culture and religion on sexual and reproductive health indicators and help-seeking amongst 1.5 generation migrants. An online survey was completed by 111 participants who answered questions about their cultural connectedness, religion, sexual and reproductive health and help-seeking. Kruskall-Wallis tests were used to analyse the data. There was no significant difference between ethnocultural groups or levels of cultural connectedness in relation to sexual and reproductive health help-seeking attitudes. The results do suggest differences between religious groups in regard to seeking help specifically from participants’ parents. Notably, participants who reported having ‘no religion’ were more likely to seek help with sexual and reproductive health matters from their parent(s). Managing cross-cultural experiences is often noted in the extant literature as a barrier to sexual and reproductive health help-seeking. However, while cultural norms of migrants’ country of origin can remain strong, it is religion that seems to have more of an impact on how 1.5 generation migrants seek help for SRH issues. The findings suggest that 1.5 generation migrants may not need to adapt their religious beliefs or practices, despite entering a new ethnocultural environment. Given that religion can play a role in the participants’ sexual and reproductive health, religious organizations are well-placed to encourage young migrants to adopt help-seeking attitudes.

## 1. Introduction

In Australia, over 27% of Australians were born overseas, and another 20% have at least one parent born overseas. Australia has also committed to the resettlement of over 12,000 new refugees and net overseas migration contributes to over 60% of Australia’s total population growth [1]. Australia thus provides a particularly rich case study of a migrant-receiving country undergoing rapid transformation. While other countries are experiencing similar changes, Australia has a comparatively rich range of visa schemes and a rapidly increasing overall intake of migrants. In Australian major cities, migrants make up a significant proportion of the population. According to the Australian Bureau of Statistics [1], cities where the migrant population is over 25% include Sydney (38.9), Perth (37.1), Melbourne (34.6), Adelaide (27.4), Brisbane (27), Darwin (25.9) and Canberra (25.3).

The cohort of interest is referred to as 1.5 generation migrants because they are not the conventional first generation migrant, who are old enough to emigrate on their own, nor are they the conventional second generation migrant, the offspring of the first generation migrant born in the new country [2].

### 1.1. The Role of Culture and Religion in Constructions of Sexual and Reproductive Health (SRH)

The cross-cultural positionality and/or religiosity of some migrants is often cited as having an impact on SRH decision-making processes [3]. Cultural and religious differences between a migrant’s country of origin and that of immigration are linked with reduced help-seeking across a range of health outcomes [4], and especially with regard to sexual and reproductive health (SRH) [5]. SRH may be of particular note as many cultures and religions have quite clear ideologies about sexuality, sexual behaviour, and thus SRH [6,7]. Given this reality, research indicates that when migrants feel bound to constructions of SRH, as per their ethnic origins or religious doctrines, they may not utilize SRH services. Migrants may perceive them to be inappropriate for their needs or that seeking such services would be perceived of negatively by their cultural or religious group (especially if strong ties are still present) thus tainting their sociocultural identity as well [7]. This type of sociocultural clash may be intensified for 1.5 generation migrants who may be culturally and/or religiously from two worlds and may thus be conflicted about how to seek help for their SRH needs while at the same time maintaining the values. 

### 1.2. Cross-Cultural and Intergenerational Understandings of SRH 

These 1.5 generation migrants not only contend with cross-cultural and religious understandings of SRH, but must also navigate intergenerational differences in the midst of cross-cultural parenting. For example, research indicates that in the first few years of arrival, first generation skilled Zimbabwean migrants found the ways in which Australian culture constructed and dealt with sexuality to be confronting and at odds with their beliefs and ways of understanding sexuality [8]. This resulted in increased avoidance of and resistance to Australian constructions of SRH delivered via Australian media and Australian people [8]. As a result, families experienced conflict when trying to educate their 1.5 generation migrant children about SRH from a Shona-Zimbabwean lens within contemporary Australia [8]. This intergenerational discrepancy may exist when the only point of reference that migrant parents have about youth sexual development is from when they themselves were youths in their country of origin. They then draw on these experiences and understandings when it becomes relevant–when they have to raise youths. Until that point, contemporary youth/teenage life in Australia or their country of origin may seem irrelevant. Furthermore, first generation migrant parents and 1.5 generation migrant children indicated that many parents of 1.5 generation children expected these children to comply with constructions of sexuality from their country of origin [8]. In addition, these expectations were more readily expressed and enforced for 1.5 generation migrant children than for second generation children/siblings born in Australia. Notable expectations include avoiding interactions with members of the opposite sex (especially enforced with girls), restrictions on participation in youth peer events (e.g., birthday parties, sleep-overs, or group excursions) and restrictions on engagement with LGBTIQ people, information, or media. 

### 1.3. Exploring SRH with 1.5 Generation Migrants

Despite the dearth of research in this area evidence indicates that 1.5 generation migrants, especially of non-Western backgrounds, often enter a new (Anglo/Euro-centric) cultural and secular environment when they move to Australia. This environment has its own set of constructs of SRH which 1.5 generation migrants are confronted with at a crucial time in their psychosexual development–childhood, puberty and adolescence [8]. This may result in having learnt and being expected to uphold (by other members of one’s cultural community) particular norms about SRH [9] from their culture of origin while at the same time adopting and enacting Australian secular constructions of SRH contributing to a culture clash [8]. Such a clash may have immediate and far-reaching implications for the SRH of 1.5 generation migrants. For migrants arriving from countries with very different cultural, ethnic and religious values, and beliefs to those in Australia the process of adapting constructions, understandings and experiences of sexuality often results in a number of challenges. This study was therefore designed to investigate the role of culture and religion on sexual and reproductive health indicators and help-seeking amongst 1.5 generation migrants.

## 2. Methods

This paper focuses on the results of the quantitative questionnaire portion of a larger project conducted in 2015. The larger project used a mixed methods cross-sectional design (i.e., quantitative questionnaire, qualitative interview and Q Methodology) to explore constructions of SRH and SRH help-seeking amongst 1.5 generation migrants in Greater Western Sydney (see [2] for results of the Q Methodology study). The Q methodology helped us to create conceptual maps of participant perspectives as it allows for the sampling of subjective viewpoints, and assists in identifying patterns, including areas of difference or overlap, across various perspectives on a given phenomenon. The Q methodology combines elements from qualitative and quantitative research traditions to understand and explore the many facets of a range of phenomena simultaneously [10]. 

Greater Western Sydney was chosen as more than 50% of its approximately 800,000 people are migrants or their descendants [1]. Furthermore, the region has been found to have pockets of cultural concentration which allows migrants to stay connected to key aspects of their culture, such as their ethnicity, community, language, and religion. To that effect, it is likely that the cultural and religious norms of migrants’ country of origin remain strong and may therefore have a significant influence on how 1.5 generation migrants in this region construct, experience, and understand various aspects of SRH. The study therefore sought to address the following questions:Do ethnicity and cultural connectedness influence 1.5 generation migrants SRH help-seeking?Does religious affiliation influence 1.5 generation migrants SRH help-seeking?From which sources are1.5 generation migrants most likely to seek SRH support?What barriers or facilitators do 1.5 generation migrants perceive to have an impact on their SRH help-seeking?

### 2.1. Survey

The survey (see Appendix A) was specifically designed for this investigation and began with demographic questions including what year the participant moved to Australia, with whom, and at what age. Participants were also asked about their religious affiliation and ethnicity. With regards to cultural connectedness, participants were asked to rank, on a 5-point Likert scale, how strongly they identified with the culture and values from their country of origin and with Australian culture. They were also asked to rank how strong relationships were with their community based on their culture of origin and the extent that cultural values created strong ties between the participant and their family. Questions on participants’ SRH history, safer sex practices, and prospective SRH help-seeking were posed. With regard to their help-seeking attitudes, participants were asked: “If you were having a sexual and reproductive health concern, how likely is it that you would seek help from the following people/places? Please indicate your response by clicking on the number that best describes your intention to seek help from each help source that is listed.” Participants then indicated on a 5-point Likert scale the likelihood of them seeking help from an intimate partner, friends, parent, other relative/family member, sexual health clinic, the Internet, a doctor/general practitioner (GP), or community/cultural or religious leader, or alternatively if they would not seek help, or would seek help from another source not listed above. Finally, participants were also asked about barriers and facilitators to seeking SRH support.

### 2.2. Participant Recruitment

A cohort of 1.5 generation migrants were recruited via advertisements posted at seven Western Sydney University campuses and surrounding off-campus venues (e.g., major shopping malls). This was done to strategically engage participants from several suburbs within the Greater Western Sydney region to ensure that the data collected were from as many ethnocultural groups as possible. Individuals over 18 years old who indicated that they had migrated as children (under 18 years old) to Australia were included in the study. No upper age limit was set as an exclusion criterion to participation.

### 2.3. Ethics Approval

This study is part of a larger research project examining the SRH of 1.5 generation migrants in Australia and ethical approval was received from the Human Research Ethics Committee of Western Sydney University. In addition, informed consent to participate in this study was obtained from all participants (approval date and code: 19 June 2015, H11168).

### 2.4. Data Analysis

Using SPSS (version 23.0. IBM, Armonk, NY, USA), quantitative data analysis software, the data were cleaned to exclude incomplete responses (x = 121) and the following analyses were run: descriptive statistics, correlations, and Kruskall-Wallis tests. Kruskall-Wallis tests were used as an alternative to one-way ANOVAs given that groups sizes were small and uneven [11]. To identify whether the salience of one’s cultural identity related to their help-seeking, Pearson product-moment correlations were performed between the measures of cultural connectedness and sources of help (Intimate Partner, Friend, Parent, Relative, Sexual Health Clinic, Internet, Doctor/general practitioner (GP), Community Leaders, No Help) using an alpha level of 0.05. As the sample was considered robust (*N* = 111), all assumptions were satisfactory. Additionally, Pearson product-moment correlations were performed between all sources of help to examine whether one help-seeking action related to another. Regarding seeking help from parents, a series of 15 post hoc pairwise comparisons were conducted using Mann-Whitney *U* tests and an adjusted alpha of 0.003. 

### 2.5. Sample Demographics

The sample consisted of 111 participants from across the Greater Western Sydney (see Table 1). The majority of participants were female (51.4%), with a nearly equal number of males (47.7%) and one participant identifying as transgender. Participants’ ages at the time of participation ranged between 16 and 60, with a mean age of 22.90 (*SD* = 5.25). Most participants were single (*n* = 82.9%) and had no children (94.6%). Seventy-six participants arrived in Australia between 2000 and 2009 (68.4%) with their close kin (mother 83.8%, father 71.2%, sibling 46.8%). The majority migrated from Sub-Saharan Africa (25%), closely followed by South-East Asia (24%), with the others migrated from East Asia (13%), the Middle East (11%), Eastern Europe (9%), the Pacific (6%), the Americas (6%), Western Europe (4%), and North Africa (2). The mean age at the time of migration was 11 years old (Mean (M) = 11.90, Standard Deviation (SD) = 4.67). The majority spoke English as a primary language (66.7%). Twenty-four languages were noted by those whose primary language was not English. The majority indicated a religious affiliation (87.4%), with 55% of those being Christian/Catholic. Ninety-five participants were heterosexual (85.5%), eight were bisexual (7.2%), five were homosexual (4.5%), one identified as lesbian (0.9%) and one identified as other (0.9%), and prefer not to say (0.9%), respectively.

## 3. Results

The present study sought to examine the role an individual’s culture has in the construction of their sexual and reproductive health. Table 2 presents the degree to which a participant’s cultural identity was determined by their cultural connectedness to their Country of Origin, Australian Culture, Community, or Family. 

The results indicate that stronger identification with one’s family positively correlates with seeking help from an intimate partner, a doctor, community leaders, and seeking no help. Table 3 depicts correlations between the measures of cultural connectedness and sources of help. Table 4 depicts correlations between the sources of help. The results indicate significant positive correlations between a strong identification with one’s country of origin and seeking help from an intimate partner, parents, a sexual health clinic, the Internet, and a doctor.

Analyses indicated significant correlations between the identification with one’s country of origin, Australian culture, one’s community, and one’s family and various sources of help, whereby stronger connections related to stronger inclinations toward seeking help from specific sources. Interestingly, seeking help from an intimate partner or doctor/general practitioner (GP) was significant across all measures of cultural connectedness. Additionally, seeking help from various sources often related to seeking help from other sources. However, stronger inclinations to seek help from a relative or sexual health clinic were significantly related to lower inclinations to seek no help.

To identify group differences between participant’s religious identifications (No Religion, Catholic/Christian, Greek Orthodox, Islamic, Buddhist, Other) among the various sources of help (Intimate Partner, Friend, Parent, Relative, Sexual Health Clinic, Internet, Doctor/GP, Community Leaders, No Help), Kruskall-Wallis nonparametric tests were conducted to accommodate the uneven group sizes. A statistically significant difference was identified for receiving help from parents (*Χ*^2^ [5, *N* = 111] = 11.30, *p* < 0.05, *η*^2^ = 1.16).

These results suggest significant differences between religious groups in regard to seeking help from parents. No significant differences, however, were found between the six religious categories—most likely due to small group sample sizes. However, the results show a significant difference only between religious affiliation and seeking help from a parent. Table 5 depicts the degree to which individuals of various religious identities seek help from their parent(s). 

The present study also sought to determine which sources individuals felt most comfortable seeking help from. Table 6 indicates participants’ perceived likelihood (in percentage) to seek help from various sources. Doctors/GP (92.7%), sexual health clinics (88.1%), the Internet (84.1%), and intimate partners (81.1%) were among the most likely sources of help, while community leaders (72.5%), relative(s) (60%), and no help (56.8%) were among the most unlikely sources of help.

The present study also sought to ascertain the most dominant barriers and facilitators to individual’s help-seeking attitudes. Among the barriers hindering individuals’ help-seeking, a lack of knowledge was identified as the most dominant barrier (45.9%). This was followed by concerns regarding concealment from one’s family and community (36.0%). These results are complimented by the facilitator of help-seeking, whereby an increase in knowledge was identified as the most dominant facilitator of help-seeking (63.1%). Similarly, assurance of concealment was identified as the second most dominant facilitator of help-seeking (45.9%). Table 7 and Table 8 depict the barriers and facilitators of help-seeking.

To contextualise the key findings, participants’ sexual and reproductive health histories were recorded. It was identified that 60.40% (*n* = 67) of the participants were currently sexually active. Of the 111 participants, 49.50% (*n* = 55) used contraceptives, 11.70% (*n* = 13) did not use contraceptives, and 38.70% (*n* = 43) preferred not to answer. Table 9 depicts the types of contraceptives participants have previously used.

With regard to prior sexual health concerns, 2.7% (*n* = 3) of participants had previously been diagnosed with an STI. Among those, 66.7% (*n* = 2) were diagnosed with gonorrhoea, while 33.30% (*n* = 1) were diagnosed with herpes. Additionally, 66.7% (*n* = 2) took antibacterial medications, while 33.3% (*n* = 1) sought help from a doctor. When queried about the duration leading to their help-seeking behaviours, it was revealed that 66.70% (*n* = 2) sought help within 1—3 days of having sex while 33.3% (*n* = 1) sought help within 4—7 days. Participants justified this by saying that they were not aware that they were infected with an STI (*n* = 2, 66.7%) and that they were hoping that the STI would go away without intervention (*n* = 1, 33.3%).

In terms of pregnancy, 9.0% (*n* = 10) had previously experienced an unplanned pregnancy. Among these participants, 40% (*n* = 4) kept the child, 40% (*n* = 4) terminated the pregnancy, 10% (*n* = 1) organised an adoption, and 10% (*n* = 1) preferred not to answer on the outcome of the pregnancy.

## 4. Discussion

This study was designed to investigate the role of culture and religion on sexual and reproductive health indicators and help-seeking attitudes amongst 1.5 generation migrants using a quantitative survey. Overall, the results suggest that 1.5 generation migrants were most likely to seek help from doctors/general practitioners (92.7%), sexual health clinics (88.1%), the Internet (84.1%), and intimate partners (81.1%) regarding clinical SRH issues. For support on non-clinical SRH matters, the results suggest that 1.5 generation migrants feel the least comfortable seeking SRH support from community leaders (72.5%) and relative(s) (60%). These findings can be further contexualised when culture and religiosity are considered.

With regards to the role of cultural connectedness on 1.5 generation migrants SRH help-seeking, the results indicate significant positive correlations between a strong identification with one’s country of origin and seeking help from an intimate partner, parents, a sexual health clinic, the Internet, and a doctor. Stronger identification with one’s family positively correlates with seeking help from an intimate partner, a doctor, community leaders, and seeking no help. This is in line with research indicating that some youths of minority and migrant backgrounds often struggle to engage with their parents when they experience an SRH concern for fear of the consequences of transgressing ethnocultural or religious protocols held in high esteem by their parents [12,13]. However, this was not the case for all of the 1.5 generation migrants in this study. This may be because these migrants feel more connected to their parents in line with their collectivist ethnocultural values [14]. For those who sought help from parents, it could also be that both the youth and their parents have acculturated more than popular discourses give them credit for [14]. 

In this study, strong identification with Australian (secular, individualist, capitalist and Eurocentric) culture positively correlates with seeking help from an intimate partner, relatives, a sexual health clinic, a doctor, and community leaders, while stronger identification with one’s community positively correlates with seeking help from an intimate partner, relatives, a doctor, community leaders, and seeking no help. Other studies highlighted that culture as a significant factor in SRH help-seeking [6]; however, the findings of this study suggest that 1.5 generation migrants are not influenced by culture to the same extent as their older counterparts [14]. These findings suggest that the colloquially perceived ethnocultural values between more recent migrants and those with a longer history in Australia are not so incongruent [14]. These findings can inform contemporary discourses about young migrants and their SRH help-seeking needs.

The study inquired about whether religious affiliation influenced 1.5 generation migrants’ SRH help-seeking. The analyses identified a significant difference only between religious affiliation and seeking help from a parent. This may be because increased religiosity has been linked to difficulties in seeking help for SRH issues from close family members due to fear of social sanctioning, as contemporary Australians youths’ sexual behaviour is often at odds with religious doctrine [2]. Notably, those with no religious affiliation were slightly more likely to seek help from parents, yet there were no statistically significant differences between the six religious affiliations. The findings therefore suggest that more inquiry is needed into the role of religiosity and SRH help-seeking amongst young migrants and culturally and linguistically diverse youth. 

To support access to SRH supports, the reduction in barriers and increase in facilitators is required. In this study, the top three barriers as perceived by 1.5 generation migrants were; not knowing where to access SRH services (45.90%), ensuring that their family and community did not find out (36.00%), and not having enough money to pay for SRH services (28.80%). Likewise, being made aware of where the services are (63.10%), being confident that no one would find out (45.90%), and access to services which are free/low cost (36.90%) were identified as the most dominant facilitators of help-seeking. These findings are aligned with Australian and international research with minority youth, aged 16 to 24, indicating that increased awareness of services that provide inconspicuous access to free SRH services improve youth SRH outcomes [15,16,17,18]. For instance, SRH support provided at university campuses can offer confidentiality from family and the community and often include billing options for local and international students that require minimal to no payment upfront [17,19,20]. However, such services are only accessible to those whose social determinants allow them the privilege of attending university. Considering that religion was an important influence in help-seeking, religious organisations may be well placed liaisons between youths, their families and communities, and SRH services. 

## 5. Limitations

The study findings reiterate the role of cultural connectedness and religiosity in SRH help-seeking for migrant youths. The study has also highlighted key areas which require further consideration and investigation. The purposeful nature of the sampling strategy helped to achieve a varied sample with the aim of capturing perspectives from various ethnic, religious, and migration backgrounds. However, the country of origin of the sample was not proportional, as most participants were from sub-Saharan Africa. In addition, the majority of participants were Catholic or Christian, which may not reflect many 1.5 generation migrants who do not prescribe to Christianity. This cultural similarity may mean the full breadth of cross-cultural SRH help-seeking perspectives and behaviours have yet to be explored. Additionally, although participants’ mean age of migration was 11 years old, those who arrived much younger may not experience as much pressure or culture clash, as they may have been too young to remember or for their families to feel that they had to adhere to the rules of their ethnic origins. The age of participants is also relevant in relation to when they migrated to Australia. For instance, as the participants aged, they may be less likely to recall or recount their experiences as children. Further, their perspectives of SRH help-seeking were asked in relation to the present versus help-seeking in the past, which would have included fewer SRH services and engagement from community services and networks. Irrespective of age at participation, it seemed that for the migrants in this study, religion appeared to hold more weight in determining their SRH help-seeking attitudes. More exploration is needed to determine the interaction between age of migration and SRH help-seeking and outcomes. Finally, the analysis was restricted, as one-way ANOVAs could not be conducted on the studies due to the small and uneven sample sizes; as such, Kruskall-Wallis tests were used instead. Ultimately, generalisations cannot be made about the different perspectives among such groups, and further study is recommended to assess the effect of diverse religious backgrounds on SRH help-seeking amongst migrants in Australia. 

Although participants of this study were recruited from a number of Western Sydney suburbs, this was done in relation to seven Western Sydney University campuses and surrounding off-campus venues (e.g., major shopping malls). As a result, the participants are likely to have been university students or staff and therefore well-educated. In such a case, the participants would potentially have a heightened capacity to both understand and critically analyse the statements before sorting them. As such, the sample may not be representative of the many 1.5 generation migrants who may not have high levels of education. With lower levels of education come lower levels of health literacy [21]. Consequently, participants’ perspectives on health care services and the engagement of these migrants with those services may be influenced by their increased ability to scrutinise, navigate, and mediate their experiences within the Australian health care system compared to other groups of migrants. Expansion of this study to include a broader variety of 1.5 generation migrants is therefore required. 

## 6. Conclusions

The influence of a cross-cultural upbringing is often noted in the extant literature as a potentially challenging factor in migrant youths’ sexual and reproductive health help-seeking. Amongst the 1.5 generation migrants in this study, there were no significant differences between ethnocultural groups or levels of cultural connectedness in relation to sexual and reproductive health help-seeking. While cultural norms of migrants’ country of origin can remain strong, it is religion that seems to have more of an impact on how 1.5 generation migrants construct, experience, understand, and engage with various aspects of SRH. The present study’s results suggest differences between religious groups in regard to seeking help specifically from youths’ parents. Notably, participants who reported having ‘no religion’ were more likely to seek help with sexual and reproductive health matters from their parents. Given that religion can play such an important role in youths’ sexual and reproductive health religious organisations may be well-placed to encourage youth help-seeking. This may be a means of addressing the barriers that youths perceive to accessing support in ways that ensure equitable and easy access to confidential and low to no cost sexual and reproductive health services.

## Figures and Tables

**Table 1 ijerph-18-01341-t001:** Demographic information for the study sample.

Demographic Information	*n*	(%)
Gender		
Male	53	47.7
Female	57	51.4
Transgender	1	0.9
Marital Status		
Single	92	82.9
De Facto	6	5.4
Married	8	7.2
Divorced	3	2.7
Engaged	1	0.9
N/A	1	0.9
Parent of Child		
Yes	6	5.4
No	105	94.6
Year of Arrival		
1960–1969	1	0.9
1970–1979	0	0
1980–1989	1	0.9
1990–1999	11	9.9
2000–2009	76	68.4
2010–2017	21	18.9
Arrived with:		
Mother	93	83.8
Father	79	71.2
Sibling	52	46.8
Grandparent	4	3.6
Aunt/Uncle	6	5.4
Extended Family	5	4.5
Family Friends	4	3.6
Alone	4	3.6
English as Primary Language		
Yes	74	66.7
No	37	33.3
Religion		
No Religion	14	12.6
Catholic/Christian	61	55.0
Greek Orthodox	4	3.6
Islamic	24	21.6
Buddhist	3	2.7
Other	5	4.5
Sexual Orientation		
Heterosexual	95	85.5
Homosexual	5	4.5
Lesbian	1	0.9
Bisexual	8	7.2
Other	1	0.9
Prefer Not to Say	1	0.9
Region of Origin		
Sub-Saharan Africa	25	24.0
North Africa	2	2.0
South East Asia	24	25.0
East Asia	13	13.0
Eastern Europe	9	9.0
Western Europe	4	4.0
Middle East	11	11.0
The Americas	6	6.0
The Pacific	6	6.0

**Table 2 ijerph-18-01341-t002:** Cultural identity in relation to participants’ perceived cultural connectedness (%).

	Strongly Agree	Agree	Neutral	Disagree	Strongly Disagree
Country of Origin	43.2	31.5	19.8	4.5	0
Australian Culture	22.5	30.6	35.1	9.0	2.7
Community	41.4	27.0	15.3	10.8	4.5
Family	49.5	31.5	12.6	3.6	1.8

**Table 3 ijerph-18-01341-t003:** Bivariate correlations between measures of cultural connectedness and sources of sexual and reproductive health (SRH) help-seeking.

	Country of Origin	Australian Culture	Community	Family
Intimate Partner	0.22 *	0.34 **	0.36 **	0.28 **
Friend	0.05	0.03	0.04	0.00
Parent	0.20 *	0.19	0.20 *	0.21
Relative	0.07	0.27 **	0.13	0.13
Sexual Health Clinic	0.32 **	0.33 **	0.12	0.12
Internet	0.20*	0.12	0.14	0.08
Doctor/GP	0.40 **	0.26 **	0.22 *	0.28 **
Community Leaders	0.10	0.22 *	0.34 **	0.23 *
No Help	0.15	−0.08	0.27 **	0.28 **

*Note*. Correlations marked with an asterisk (*) and double asterisk (**) were significant at *p* < 0.05 and *p* < 0.01, respectively.

**Table 4 ijerph-18-01341-t004:** Bivariate correlations between sources of SRH help-seeking.

	1.	2.	3.	4.	5.	6.	7.	8.
1. Intimate Partner	—							
2. Friend	0.22 *	—						
3. Parent	0.24 *	0.15	—					
4. Relative	0.28 **	0.14	0.50 **	—				
5. Sexual Health Clinic	0.45 **	0.10	0.30 **	0.23 *	—			
6. Internet	0.34 **	1.8	−0.18	−0.10	0.20 *	—		
7. Doctor/GP	0.32 **	−0.03	0.25 **	0.04	0.60 **	0.14	—	
8. Community Leaders	0.11	−0.02	0.46 **	0.44 **	0.11	−0.08	0.17	—
9. No Help	−0.09	−0.13	−0.19	−0.23 *	−0.37 **	−0.01	−0.19	0.16

*Note*. Correlations marked with an asterisk (*) and double asterisk (**) were significant at *p* < 0.05 and *p* < 0.01, respectively.

**Table 5 ijerph-18-01341-t005:** Participants’ perceived likelihood of SRH help-seeking from parent among religious identities.

	Parent	
Religious Identity	*M*	*SD*
No Religion (*N* = 14)	4.00	0.88
Catholic/Christian (*N* = 59)	3.07	1.30
Greek Orthodox (*N* = 4)	3.50	1.00
Islamic (*N* = 24)	3.54	1.29
Buddhist (*N* = 3)	3.67	0.58
Other (*N* = 5)	2.20	1.10

**Table 6 ijerph-18-01341-t006:** Perceived likelihood (%) of SRH help-seeking depending on source.

	Extremely Likely	Likely	Neutral	Unlikely	Extremely Unlikely
Intimate Partner	52.3	28.8	8.1	7.2	3.6
Friend	12.7	34.5	24.5	20.9	7.3
Parent	11.0	16.5	24.8	28.4	19.3
Relative	3.6	11.8	24.5	29.1	30.9
Sexual Health Clinic	52.3	35.8	6.4	4.6	0.9
Internet	57.0	27.1	8.4	1.9	5.6
Doctor/GP	61.5	31.2	5.5	1.8	0
Community Leaders	2.8	3.7	21.1	25.7	46.8
No Help	3.7	6.4	33.0	28.4	28.4

**Table 7 ijerph-18-01341-t007:** Participants perceptions of potential barriers to SRH help-seeking.

	*n*	(%)
I don’t know where these services are	51	45.9
The risk that my family/community could possibly find out	40	36.0
These services do not cater well to people of my ethnicity/culture	15	13.5
These services cost too much money	32	28.8
These services are too far away from where I live	12	10.8
Service trading hours	16	14.4
I have other ways of getting support/assistance	16	14.4
Other	2	1.8

**Table 8 ijerph-18-01341-t008:** Participants perceptions of potential facilitators of SRH help-seeking.

	*n*	(%)
Being made aware of where the services are	70	63.1
Being confident that no one would find out	51	45.9
Knowing that there are health workers who cater towards my ethnicity/culture	26	23.4
Services which are free/low cost	41	36.9
Services which are close to where I live	36	32.4
Trading hours which include evenings/weekends	31	27.9

**Table 9 ijerph-18-01341-t009:** Types of contraceptives used by 1.5 generation migrants in Australia.

	*n*	(%)
Condoms	51	45.9
Birth Control Pills	26	23.4
Diaphragm	4	3.6
Intrauterine Device (IUD)	1	0.9
Vaginal Ring	0	0
Implant	1	0.9
Patch	1	0.9
Emergency Contraception	8	7.2
Permanent	0	0

## Data Availability

The data presented in this study are available on request from the corresponding author. The data are not publicly available due to ethical restrictions on public access.

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
