# Peer review of "The Role of Culture and Religion on Sexual and Reproductive Health Indicators and Help-Seeking Attitudes amongst 1.5 Generation Migrants in Australia: A Quantitative Pilot Study"

_ijerph, 2021, doi:10.3390/ijerph18031341_

Round 1

Reviewer 1 Report

Dear Editors and Authors

I want to thank the authors and the Editor of Int. J. Environ. Res. Public Health for the opportunity to review this interesting manuscript.

Attached are my suggestions divided into the sections in which they appear, with reference to line number alternatively the “sub-section” of the paper.

Author Response

Abstract:

Line 23: The sentence starts with a number. Is it possible to rephrase so that the sentence does not start with a number?

  • Revised.

Lines 27‐29 says: “The results do suggest differences between religious groups in regards to seeking help specifically from young peoples’ parents. Notably, youth who reported having ‘no religion’ were more likely to seek help with sexual and reproductive health matters from their parent(s).” Are all participants in the study considered to be “young people” or “youths” or could these words be replaced with “the participants”?

  • Revised.

Introduction:

Lines 51‐54: Is there a reference to the definition of the 1.5 generation migrants that could be added to this sentence?

  • Citation added.

Line 98 could the “SRH‐ a culture clash” be replaced with “SRH contributing to a culture clash” or something along those lines?

  • Revised.

Methods:

Line 106: Somewhere here it would be helpful if the authors could briefly mention (if space allows) what Q Methodology stands for. If it is possible to describe in a few words.

  • Additional information on Q Methodology has been added. There is no information on what the Q in Q Methodology stands for.

Line 106: Was it “qualitative research interviews” that were conducted? If yes, perhaps this should be added.

  • Revised.

First section below methods: This section is somewhat of a mix of methods and setting. I wonder if the words “this paper will focus on quantitative survey of the larger project” (now similar to what is on line 114‐115) could be introduced after the first sentence. Then it could also be helpful to use the same terminology in the first sentence (line 105‐106). I.e. that the authors either refer to a survey or a questionnaire throughout. If this revision is made then the new sentence on lines 114‐115 could say “The study addressed (or sought to address) the following research questions”.

  • Revised.

Line 118: This research question states: “From which sources do 1.5 generation migrants feel most comfortable seeking SRH support?” While the question in the survey seems to have asked from which source, the participants are most likely to seek help. I wonder if this research question can be reformulated so that it is closer to the question asked?

  • Revised.

General comment: Is it possible to add which year the current study was conducted?

  • The year (2015) has been added.

Section “2.1 Survey”: Is it possible to add the survey as an appendix or additional file?

  • The survey has been included as an additional file on the IJERPH portal.

Line 136: Is it possible to add the words “alternatively if they” before the words “would not seek or would seek help from another source not listed above” in the sentence?

  • Revised.

Section “2.2. Participant recruitment”: Do the authors know how large of the sample that was recruited via the university campus in comparison to the “off‐campuses” venues?

  • No, this data was not collected from participants however the proximity to university campuses and the likelihood of specific types of participants is noted as a limitation in that section.

General question: Did the author discuss any upper age limit for inclusion in the study? If no, could there be any potential limitations to the inclusion of older participants?

  • The following has been added to the paper; “No upper age limit was set as an exclusion criterion to participation”. Only 2 participants arrived to Australia between 1960 and 1989 which does not have a significant impact on the findings. This has nonetheless been indicated in the limitations section.

Data analysis: Please add which version of the SPSS that was used for the analysis.

  • Revised.

Results:

General comment: I would suggest the authors to review so that all percentages presented in the manuscript either have one or two decimals. I would suggest using only one decimal.

  • Revised throughout.

General comment: It would be helpful if the authors could guide the readers to the results by stating one or two sentences in relation to each table summarizing what the main results (or take home message) for each table is. This is done for some of the tables but not all.

  • Revised for Tables 2, 3 and 5.

Table 2: Is it possible to revise the table text slightly for further clarification? Also so that the table text is able to be read independently from the text. Now the table heading says “degree of cultural connection” and the response alternatives ranges from strongly agree to strongly disagree while them text refers to “cultural identity in relation to their cultural connectedness”.

  • Table titles have been revised.

Lines 178‐181: Does this belong to the methods (analysis)?

  • Revised all sentences referring to forms of analysis have been moved to the analysis section of the paper.

Line 211‐212: This refers to the previous comment on the third research question. Should this sentence be reformulated so that it refers to “which sources the participants are most likely to seek help from” instead of which sources participants felt most comfortable seeking help from.

  • Revised as noted above.

Table 6: Should the table text say “Perceived likelihood of help seeking depending on source” or something along those lines.

  • Revised as suggested.

Discussion:

General comment: A reminder to check so that the percentages are consistently presented with either one or two decimals.

  • Revised throughout.

Line 250: See formulation regarding “feeling comfortable” as have been addressed also in previous comments.

  • Revised.

Lines 290‐298: These lines discussed the youth perspective. I wonder if the authors would like to address how reflective this is in relation to the age range and mean age of the included participants in the current study? The same applies to the last lines of the conclusions (lines 339‐343). I would also suggest the authors to consider choosing either “young people” or “youth” if possible, for consistency if referring to this in the discussion (and the conclusion).

  • The addition of “aged 16 to 24” has been added to specify the age range to which we are referring. Given that the mean age of participants was 22.90 which falls within customarily identified youth age ranges.
  • Youth has been used throughout the paper.

Reviewer 2 Report

Dear Authors, 

Thank you for well-prepared actual research. It was attractive for me and clearly presented. If I can suggest something - I know, that the current project is only part of huge survey, but your further ideas and ways of developing could be interested. Also, I do not know about other studies that might be similar to yours. Have you some information concerning other? 

Author Response

Thank you for well-prepared actual research. It was attractive for me and clearly presented. If I can suggest something - I know, that the current project is only part of huge survey, but your further ideas and ways of developing could be interested.

  • These are already included in the discussion sections.

Also, I do not know about other studies that might be similar to yours. Have you some information concerning other? 

  • We have cited relevant articles in the background, discussion and conclusion sections.

Reviewer 3 Report

Overall, the manuscript was well written and would contribute to the literature on refugee studies.

There are a few minor issues:

Page 1 in Line 42: 12.000 should be 12,000.

Page 3 in Line 114: "will focus" should be "focus" or "focused."

Author Response

Page 1 in Line 42: 12.000 should be 12,000.

  • Revised.

Page 3 in Line 114: "will focus" should be "focus" or "focused."      

  • Revised.

This sentence is awkward and needs to be reworded:

“This suggests that while 1.5 generation migrants may need to adapt to a new ethnocultural environment little about their religious beliefs or practices may require adaptation in Australia.”

Revised

In the following sentence, there might be a comma (,) after “…reproductive health.”

“Given that religion can play a role in young peoples’ sexual and reproductive health, religious organisations are well placed to encourage young people’s help-seeking attitudes.”

Revised

Page 4: Kruskall-Wallist test is non-parametric test equaivalent to one-way ANOVA. Namely, when the assumptions of one-way ANOVA do not meet (i.e., normal distribution of dependent variables and equal variances among more than two groups), Kruskall-Wallist test is used. So I believe the authors need to change one-way ANOVA instead of MANOVA in Page 4 (Line 153). If the authors need to keep MANOVA (which is used when multiple dependent variables are treated as one dimension), please explain why Kruskall-Wallist test is an alternative method for MANOVA rather than one-way ANOVA.

  • We have changed line 160 in the revised manuscript from MANOVA to one-way ANOVA
  • This has also been changed in the Limitations on line 331

The authors noted they used Kruskall-Wallis tests due to the small and uneven sample size. As just noted, Kruskall-Wallis test is used when normality assumption and equal variance assumption among groups are violated. The uneven sample size may lead to non-equal variance assumption among the groups. So if the dependent variables are not assumed to normally distributed, please note non-normality assumption as a reason to use Kruskall-Wallis test instead of parametric tests such as one-way ANOVA. Also, if the uneven sample size likely yields non-equal variance among the groups, please note non-equal variance among the groups as a result of the small and uneven sample size.

  • We have revised in the manuscript (line 159-160) Kruskall-Wallis tests were used as an alternative to one-way ANOVAs given that groups sizes were small and uneven
  • The following reference has also been added (Hills, A. (2014). Foolproof Guide To Statistics Using IBM SPSS (Custom Edition): Pearson Australia Pty Limited.)

Page 6: The authors used Pearson-Product-moment correlation. Pearson-Product-moment correlation is used when the two variables are normally distributed. Please note the two variables can be assumed to be normally distributed. If the authors found the two variables are seriously violating normality assumption, I would suggest the authors conduct Spearman Rho or Kendal’s tau correlation coefficient.

  • We have left this aspect as is because Pearson Product moment correlations were conducted on the entire sample (n = 111), not each religious group (as was the case with the above). Assumptions of normality and homogeneity were met and we were able to conduct the Pearson Product moment correlation analyses.